# Impact of Discontinuing Levofloxacin Prophylaxis on Bloodstream Infections in Neutropenic Hematopoietic Stem Cell Transplantation Patients

**DOI:** 10.3390/antibiotics11091269

**Published:** 2022-09-19

**Authors:** Thaís Guimarães, Igor Carmo Borges, Fernanda de Souza Spadão, Livia Mariano, Marina de Mattos Nascimento, Hermes Higashino, Flavia Rossi, Vanderson Rocha, Silvia Figueiredo Costa

**Affiliations:** 1Department of Infection Control, Instituto Central, Hospital das Clínicas, University of São Paulo, São Paulo 05508-220, Brazil; 2Infectious Diseases Department, Hospital das Clínicas, University of São Paulo, São Paulo 05508-220, Brazil; 3Hematology Department, Hospital das Clínicas, University of São Paulo, São Paulo 05508-220, Brazil; 4Microbiology Laboratory, Central Laboratory Division, Hospital das Clínicas, University of São Paulo, São Paulo 05508-220, Brazil

**Keywords:** hematopoietic stem cell transplantation, bloodstream infection, neutropenia, bacterial resistance, fluoroquinolone

## Abstract

Multidrug-resistant pathogens have emerged worldwide. We have driven the hypothesis that the non-use of fluoroquinolone prophylaxis during neutropenia could reduce antibiotic resistance in Gram-negative bacteria that cause bloodstream infections (BSIs) in hematopoietic stem cell transplantation (HSCT) patients and that this change in resistance pattern could lead to an impact on BSI mortality. This is a quasi-experimental study comparing BSI incidence, resistance patterns of bacteria that cause BSI, and BSI mortality when levofloxacin prophylaxis was routine for neutropenic HSCT patients (2016–2018) to when fluoroquinolone prophylaxis was discontinued in our center (2019). Bivariate comparisons and multivariate logistic regression models were used for analyses. A total of 310 HSCTs (66 (21%) allogeneic and 244 (79%) autologous) were performed during the study period. Sixty (19%) patients had BSIs, 30 in each evaluated period. The discontinuation of levofloxacin prophylaxis was associated with an increase in BSI incidence and a decrease in the resistance rates of causative BSI bacteria and in BSI 30-day mortality. The increase in the rate of resistant bacteria causing BSI and in BSI mortality might outweigh the benefits of a decrease in BSI incidence caused by fluoroquinolone prophylaxis in neutropenic HSCT patients. We suggest that the routine use of fluoroquinolone in this context be revisited.

## 1. Introduction

Bloodstream infection (BSI) is the most common serious infectious complication in patients undergoing hematopoietic stem cell transplantation (HSCT). Depending on the protocol used for transplantation and the duration of neutropenia, 13–62% of patients develop BSIs, which can result in an increased length of hospital stay, costs, and mortality [1,2,3].

Changes in the care of HSCT recipients have been shown to impact BSI etiology, especially regarding microorganism resistance patterns [4]. Although fluoroquinolone (FQ) prophylaxis has been proven to reduce the rate of BSI during neutropenia, the increasing rate of BSI due to fluoroquinolone-resistant and extended-spectrum b-lactamase (ESBL)-producing Gram-negative bacteria is of growing concern [5], and multidrug-resistant Gram-negative pathogens have emerged worldwide [6]. These epidemiologic trends are important to consider when choosing an empirical antibiotic treatment because inadequate coverage is associated with increased mortality [7].

We have driven the hypothesis that the non-use of universal levofloxacin prophylaxis during neutropenia could reduce antibiotic resistance in Gram-negative bacteria that cause BSIs in HSCT patients and that this change in the resistance pattern could lead to an impact on BSI mortality.

## 2. Results

A total of 310 HSCTs were performed during the study period (222 (72%) from 2016 to 2018 and 88 (28%) in 2019), among which, 66 (21%) were allogeneic and 244 (79%) were autologous. The median (IQR) age of the patients was 54 (38–62) years, and 167 (54%) were male. The most common underlying diseases of the patients were multiple myeloma (*n* = 107 (35%)), non-Hodgkin lymphoma (*n* = 50 (16%)), and Hodgkin lymphoma (*n* = 40 (13%)) (Table 1). Although BSI related to central venous catheter (CVC-BSI) incidence density decreased from 2016 to 2018 (pre-period; levofloxacin prophylaxis), the density of both CVC-BSI and BSI related to mucosal barrier injury (MBL-BSI) incidence increased in 2019 (post-period; non-levofloxacin prophylaxis) compared to 2018 (Figure 1). Levofloxacin prophylaxis (pre-period; 2016–2018) was associated with a decrease in BSI after adjustment for gender, underlying disease, and duration of neutropenia in the multivariate regression model (Appendix A). On the other hand, not using levofloxacin prophylaxis (post-period; 2019) was associated with a decrease in death during hospitalization after adjustment for HSCT modality, underlying disease, duration of neutropenia, and BSI occurrence (Appendix A).

A total of 60 patients (19%) had BSIs during neutropenia, of which, 31 (52%) were MBL-BSI and 29 (48%) were CVC-BSI. Among the 222 patients that started the neutropenic period from 2016 to 2018 (pre-period; levofloxacin prophylaxis), 30 patients (14%) had a BSI during neutropenia, while among the 88 patients that started the neutropenic period in 2019 (post-period; non-levofloxacin prophylaxis), 30 patients (34%) had a BSI during neutropenia (*p* < 0.001 for the bivariate analysis comparing the two periods). The comparison of BSI characteristics between the two periods is shown in Table 2. The overall BSI characteristics were similar between the compared periods, with the exception that lymphoma was a more common underlying condition in 2016–2018 (pre-period; levofloxacin prophylaxis) (60%) than in 2019 (33%).

Blood cultures identified Gram-negative bacteria in forty-four (73%) cases (three cases had two different Gram-negative bacteria), while Gram-positive bacteria were identified in eighteen (30%) cases and yeasts in one (2%) case. Three (5%) BSIs were caused by both Gram-negative and Gram-positive bacteria. There were no significant differences between the distribution of the microorganisms identified as causative agents of BSI in 2019 (post-period; non-levofloxacin prophylaxis) compared to 2016–2018 (pre-period; levofloxacin prophylaxis) (Table 3). On the other hand, BSIs from the pre-period in 2016–2018 (levofloxacin prophylaxis) were more frequently caused by Gram-negative bacteria resistant to quinolones (60% vs. 17%) and third-generation cephalosporins (43% vs. 13%) compared to the post-period non-levofloxacin prophylaxis in 2019 (Table 2). Among the 17 patients with resistance to third-generation cephalosporins, 7 (41%) were resistant due to ESBL production alone and 10 (59%) had concomitant carbapenem resistance and might have had different resistance mechanisms to cephalosporins.

Among patients with BSI during neutropenia, there was a total of 9/60 deaths (15%). All deaths occurred within 30 days after the positive blood culture, with a median (IQR) of 10 (3–17) days after blood culture collection. The BSI 30-day mortality rate was 27% (8/30) in the pre-period (2016 to 2018; levofloxacin prophylaxis) and 3% (1/30) in the post-period (2019; non-levofloxacin prophylaxis) (*p* = 0.03 for the bivariate analysis comparing the two periods). Not using levofloxacin prophylaxis (post-period; 2019) remained significantly associated with a decrease in 30-day mortality of BSI compared to levofloxacin prophylaxis use (pre-period; 2016–2018) after adjustment for the age of the patients, type of BSI (MBL-BSI vs. CVC-BSI), HSCT modality, underlying disease, and antibiotic resistance of the BSI causative agent with the inclusion of each of these possible confounding factors in separate regression models (Appendix A). However, there was a trend toward a significant association between not using levofloxacin prophylaxis and decreased BSI 30-day mortality after adjustment for HSCT modality and antibiotic resistance of the BSI causative agent in the multivariate regression model, including all independent variables associated with the outcome (Appendix A).

## 3. Discussion

In our study, we observed that the discontinuation of levofloxacin prophylaxis in neutropenic HSCT patients was associated with an increase in BSI incidence. On the other hand, the discontinuation of prophylaxis was associated with a decrease in resistance rates of causative BSI bacteria and in BSI 30-day mortality.

The decrease in BSI incidence associated with FQ prophylaxis in our study has been extensively demonstrated before, and this is the reason why FQ preventive treatment has been the standard of practice in this context [8]. Mikulska et al. (2018) reviewed two randomized controlled trials and twelve observational studies to evaluate the efficacy of FQ prophylaxis during neutropenia in patients with hematological malignancies or following HSCT [9]. FQ prophylaxis was associated with a lower rate of BSI (pooled OR 0.57, 95% CI 0.43–0.74) and episodes of fever during neutropenia (pooled OR 0.32, 95% CI 0.20–0.50). In addition, no effect of the background rate of FQ resistance was observed on the efficacy of FQ prophylaxis for settings with an FQ resistance rate below 27% [9]. 

The association we have found between the increase in antibiotic resistance among BSI causative agents and FQ preventive treatment has also been shown by other authors for colonization or infection by FQ- or multidrug-resistant bacteria [10,11,12]. A recent study has shown that FQ prophylaxis in HSCT patients was associated with breakthrough bacteremia with meropenem-resistant *Pseudomonas aeruginosa* strains, likely due to mutations increasing efflux pump activity [12]. These findings raise concern for an aggravation of the current increase in microbial resistance among HSCT patients [6].

Our finding of an increase in BSI mortality and death during hospitalization (which we believe was mainly driven by BSI-related death) associated with FQ preventive treatment contrasts with the meta-analysis of older studies by Gafter-Gvili et al. (2012) that demonstrated a decrease in infection-related death in afebrile neutropenic patients following chemotherapy who received FQ prophylaxis [13]. However, a more recent study by Henig et al. (2020) showed that the discontinuation of FQ prophylaxis in allogeneic HSCT patients led to a non-statistically significant decrease in Gram-negative BSI 30-day (35% vs. 24%) and 90-day (52% vs. 35%) mortality [14]. To our knowledge, the present study is the first to demonstrate a significant decrease in BSI mortality in neutropenic HSCT patients that did not use FQ prophylaxis compared to patients that used FQ prophylaxis. We hypothesized that the increased BSI mortality in patients that used FQ prophylaxis could be associated with the increased frequency of resistant bacteria causing BSI in that group, resulting in the inappropriateness of empiric antibiotic therapy for febrile neutropenia [15]. Although our regression analyses demonstrated that FQ prophylaxis was an independent risk factor for BSI mortality after control for antibiotic resistance, we believe that the adjustments by the multivariate regressions could have been impaired by the restricted sample size of the study. An alternative explanation for the increase in BSI mortality associated with FQ prophylaxis could be related to the selection of more virulent Gram-negative bacteria colonizing the gastrointestinal tract of HSCT patients that used FQ. That, in turn, could have resulted in BSI caused by more virulent bacterial strains compared to HSCT patients that did not receive FQ prophylaxis. Corroborating this hypothesis, it has been previously demonstrated that ciprofloxacin treatment can select more virulent strains of *Salmonella typhimurium* in the gastrointestinal tract in an animal model [16,17].

The main limitations of our study are the restricted sample size and the availability of data from only one year after the discontinuation of FQ prophylaxis. In addition, the observational design of the study might increase the risk of bias in the analyses.

In conclusion, the increase in the rate of resistant bacteria causing BSI and in BSI mortality might outweigh the benefits of a decrease in BSI incidence associated with FQ prophylaxis in neutropenic HSCT patients. We suggest that the routine use of FQ in this context be revisited. The findings of this study need to be further evaluated in large, randomized trials.

## 4. Materials and Methods

This is a quasi-experimental (before–after) study comparing the use of levofloxacin prophylaxis (pre-period) with no prophylaxis (post-period) during neutropenia in HSCT patients at Hospital das Clínicas of University of São Paulo, a teaching hospital that is a reference center for HSCT in Brazil. Levofloxacin was routinely used for prophylaxis in our institution from day 1 of HSCT to engraftment (afebrile patients) and/or the administration of an empirical antibiotic for febrile neutropenia until December 2018. In January 2019, there was a change in the institutional protocol, and levofloxacin prophylaxis during neutropenia was discontinued. There were no other changes in the institutional protocol of HSCT at that time, such as the use of piperacillin–tazobactam for the antibiotic of choice for empiric therapy in febrile neutropenia, the treatment of BSI for a minimum of 7 to 14 days, and the routine use of prophylactic antifungal therapy. We compared the incidence density of BSI and the pattern of susceptibility of Gram-negative microorganisms that cause BSI in HSCT patients between 2016 and 2018, and 2019 in our center. We evaluated the incidence density of CVC-BSI from January 2016 to December 2019 using patients-day as the denominator and MBL-BSI from 2018 to 2019 using neutropenia-day as the denominator.

### 4.1. BSI Definitions

CVC-BSI was defined as the growth of a non-skin commensal pathogen in one or more venipuncture blood cultures or the growth of a skin commensal pathogen in two or more blood cultures collected from different peripheral venopunctions of the same species and with the same susceptibility profile of the pathogen isolated from the catheter tip culture (>15 CFU with a “semiquantitative” technique) or from the blood culture collected through the lumen of central venous access with growth occurring at least 120 minutes faster in the central sample than in the peripheral sample [18].

MBL-BSI was defined as the growth of any typical intestinal microorganisms (*Bacteroides spp*, *Candida spp*, *Clostridium spp*, *Enterococcus spp*, *Streptococcus viridans*, or *Enterobacteriaceae*) in at least one blood culture sample. In addition, the patient had to meet at least one of the following criteria: (a) allogeneic bone marrow transplant within one year with one of the following documented aspects during the same hospitalization that the positive blood culture was collected: gastrointestinal graft versus host disease (GVHD) grade III or IV or ≥1 liter of diarrhea in 24 hours beginning on or within 7 calendar days prior to the date of the collection of the positive blood culture; (b) a neutropenic patient, defined as at least two different days with absolute neutrophil count values less than 500 cells/mm³ within a 7-day period of the date of the collection of the positive blood culture [17].

### 4.2. Microbiological Definitions

The microbiology laboratory used the automated system Bactec® to process blood cultures. The identification of the bacteria was performed using MALDI-TOF (Biomerieux, Craponne, France), and antimicrobial susceptibility testing was performed by automated micro dilution using a Vitek-2 (Biomerieux, France) in accordance with the current Clinical and Laboratory Standards Institute criteria at that time point [19].

### 4.3. Statistical Analysis

Data from January 2016 to December 2018 (when levofloxacin prophylaxis was routine) were compared with data from January 2019 to December 2019 (when no bacterial prophylaxis was used). The main outcome of the study was BSI 30-day mortality, defined as death by any cause within 30 days after the collection of the blood culture that identified a plausible cause of BSI. Categorical variables were presented as absolute numbers and percentages and continuous variables as medians (interquartile range (IQR)) due to their non-normal distributions. Bivariate comparisons of categorical data used the chi-square test or Fisher exact test, and continuous variables were compared using the Mann–Whitney U test. In addition, logistic regression models with 30-day mortality as the dependent variable were fitted in order to evaluate the effect of possible confounding factors on the association between the use of quinolone prophylaxis and death. Since there was a relatively small sample size with few fatal outcomes, we fitted a separate logistic regression model for each confounding factor evaluated and a multivariate regression model including all independent variables that showed a *p* value < 0.1 on the bivariate logistic regression analysis. We also used multivariate logistic regression models to evaluate factors associated with BSI and death during the hospitalization of neutropenia onset. Likewise, we included all independent variables that showed a *p* value < 0.1 on the bivariate logistic regression models in these multivariate analyses, forcing levofloxacin prophylaxis in the final multivariate regression model since it was the main independent variable of interest. Statistical tests were two-tailed with a significance level of 0.05. The software STATA (version 13.0) (Statacorp, College Station, TX, USA) was used for the analyses.

## Figures and Tables

**Figure 1 antibiotics-11-01269-f001:**
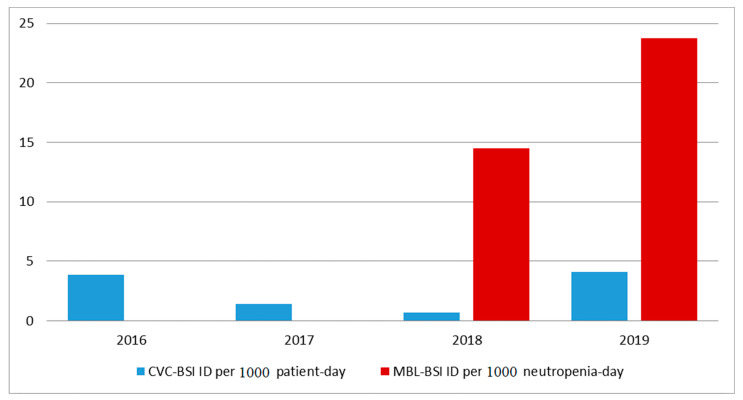
Incidence density (ID) of bloodstream infections (BSIs) related to central venous catheter (CVC-BSI) or to mucosal barrier injury (MBL-BSI) in neutropenic hematopoietic stem cell transplantation (HSCT) patients at a reference center in São Paulo, Brazil, from 2016 to 2019.

**Table 1 antibiotics-11-01269-t001:** Comparison of the characteristics of neutropenic patients undergoing hematopoietic stem cell transplantation (HSCT) before (January 2016–December 2018) and after ((January 2019–December 2019) suspension of levofloxacin prophylaxis during neutropenia.

Characteristics	Proportion (%) or Median (IQR)		*p*-Value
	January 2016–December 2018 (*n* = 222)	January 2019–December 2019 (*n* = 88)	
Male gender	121 (55)	46 (52)	0.72
Age (years)	55 (40–62)	49 (34–61)	0.11
Duration of neutropenia (days)	8 (6–10)	8 (6–11)	0.28
Underlying disease ^a^			
Multiple myeloma	73 (33)	34 (39)	0.34
Lymphoma	67 (30)	23 (26)	0.48
Leukemia	21 (10)	13 (15)	0.18
Others	61 (27)	18 (20)	0.20
HSCT type			0.01
Allogeneic	39 (18)	27 (31)	
Autologous	183 (82)	61 (69)	
Bloodstream infection	30 (14)	30 (34)	<0.001
Death during hospitalization	14 (6)	3 (3)	0.41

^a^ Each group of underlying diseases was analyzed in a separate statistical test.

**Table 2 antibiotics-11-01269-t002:** Comparison of the characteristics of bloodstream infections (BSIs) in neutropenic patients undergoing hematopoietic stem cell transplantation (HSCT) before (January 2016–December 2018) and after (January 2019–December 2019) suspension of levofloxacin prophylaxis during neutropenia.

Characteristics	Proportion (%) or Median (IQR)		*p*-Value
	January 2016–December 2018 (*n* = 30)	January 2019–December 2019 (*n* = 30)	
Male gender	19 (63)	21 (70)	0.58
Age (years)	53 (32–59)	50 (33–60)	0.59
Duration of neutropenia (days)	10 (7–13)	8 (7–11)	0.16
Fever during BSI	18 (60)	24 (80)	0.09
Duration of fever	2 (2–4)	2 (1–4)	0.34
Underlying disease ^a^			
Multiple myeloma	6 (20)	12 (40)	0.09
Lymphoma	18 (60)	10 (33)	0.04
Leukemia	5 (17)	2 (7)	0.42
Others ^b^	1 (3)	6 (20)	0.25
HSCT type			0.33
Allogeneic	8 (27)	4 (13)	
Autologous	22 (73)	26 (87)	
BSI type			0.20
MBL ^c^	13 (43)	18 (60)	
CVC ^d^	17 (57)	12 (40)	
Death within 30 days of blood culture collection	8 (27)	1 (3)	0.03

^a^ Each group of underlying diseases was analyzed in a separate statistical test. ^b^ Other underlying diseases included solid tumors (*n* = 5), amyloidosis (*n* = 1), and blastic plasmacytoid dendritic cell neoplasm (*n* = 1). ^c^ Bloodstream infection related to mucosal barrier injury. ^d^ Bloodstream infection related to central venous catheter.

**Table 3 antibiotics-11-01269-t003:** Causative agents of bloodstream infections (BSI) in neutropenic patients undergoing hematopoietic stem cell transplantation (HSCT) before (January 2016–December 2018) and after (January 2019–December 2019) suspension of levofloxacin prophylaxis during neutropenia.

Causative Agents	*n* (%)		*p*-Value
	January 2016–December2018 (*n* = 30 BSI Cases) ^a^	January 2019–December 2019 (*n* = 30 BSI Cases) ^a^	
**Gram-negative bacteria**			
Enterobacterales	20 (67)	22 (73)	0.58
Non-fermentative	2 (7)	2 (7)	1.00
Other Gram-negative bacteria ^b^	1 (3)	0	1.00
Gram-negative resistant to quinolones	18 (60)	5 (17)	0.001
Gram-negative resistant to third-generation cephalosporins	13 (43)	4 (13)	0.02
Gram-negative resistant to carbapenems	7 (23)	3 (10)	0.30
**Gram-positive bacteria**			
Coagulase-negative *Staphylococcus*	4 (13)	6 (20)	0.73
*Streptococcus* spp.	3 (10)	2 (7)	1.00
Other Gram-positive bacteria ^c^	2 (7)	1 (3)	1.00
**Yeasts**			
*Candida krusei*	1 (3)	0	1.00

^a^ Some BSI cases had identification of more than one bacterium on blood cultures. ^b^ Other Gram-negative bacteria comprised *Capnocytophaga sp*. (*n* = 1). ^c^ Other Gram-positive bacteria included *Staphylococcus aureus* (*n* = 1), *Enterococcus faecium* (*n* = 1), and *Rothia mucilaginosa* (*n* = 1).

## Data Availability

Not applicable.

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
