# Peer review of "Impact of Discontinuing Levofloxacin Prophylaxis on Bloodstream Infections in Neutropenic Hematopoietic Stem Cell Transplantation Patients"

_antibiotics, 2022, doi:10.3390/antibiotics11091269_

Round 1

Reviewer 1 Report

The authors of the manuscript entitled "Impact of discontinuing levofloxacin prophylaxis on bloodstream infections in neutropenic hematopoietic stem cell transplantation patients" presented to me a very interesting issue, linking the use of antibacterial therapy with an increase in mortality among neutropenic patients.

The authors put forward an interesting hypothesis, but the results confirming it are based on an extremely limited study group authors also had only data from one year after discontinuation of FQ prophylaxis. Despite these limitations, the study was conducted thoughtfully, and the results contained therein are presented in a comprehensible manner and well discussed. The authors' right remark is that in order to be able to confirm 100% of the observations, it would be necessary to obtain a larger study group. This work also has some editorial flaws, the authors in the resulting part introduce numerous abbreviations that do not always have their development on the first use. It is also worth noting that Figure 1 is not very precise, it requires a more detailed description of the legend and a better presentation of the data with the description of the axis. The authors also rely on a very limited number of articles, so it would be worthwhile to expand the number of cited works; thus perhaps it would be possible to enrich the discussion.

Author Response

Response: We thank the reviewer for the kind comments and suggestions.

  • We have corrected the editorial flaws related to the explanation of the abbreviations throughout the manuscript.
  • We have reformulated Figure 1 for a more detailed description of the data.
  • We have cited more articles for the enrichment of the discussion, such as “Gafter-Gvili A, Fraser A, Paul M, et al. Antibiotic prophylaxis for bacterial infections in afebrile neutropenic patients following chemotherapy. Cochrane Database Syst Rev. 2012;1:CD004386”, “Henig I, Henig O, Bar-Yoseph H, et al. Omitting Fluoroquinolones Antibiotic Prophylaxis in Allogeneic Hema-topoietic Stem Cell Transplantation Does Not Increase Gram-Negative Bacteremia Rate or Transplant-Related Mortality. Blood. 2020;136: 34-35”, and “Kadri SS, Lai YL, Warner S, Strich JR, et al. Inappropriate empirical antibiotic therapy for bloodstream infections based on discordant in-vitro susceptibilities: a retrospective cohort analysis of prevalence, predictors, and mortality risk in US hospitals. Lancet Infect Dis. 2021;21:241-251”. The discussion of the mentioned studies can be found between lines 160 – 170.

Reviewer 2 Report

This is an interesting manuscript assessing the management of HSCT patients without levofloxacin prophylaxis and more specifically, the effect on BSI occurrence, severity, and antimicrobial resistance. I have some comments that could be of use:

1.       Abstract: FQ – please define all abbreviations when first used

2.       Line 35: Change ‘problem’ to ‘condition’

3.       Figure 1: What is MBL? All abbreviations should be explained when first introduced in the abstract, the main text, the figures and the tables

4.       You could add the data on BSI mortality in Table 1 to allow the reader to easily find this information visually since this is an important finding in the study

5.       I am skeptical about this type of logistic regression analysis. I would personally find it more reasonable to perform a multivariate logistic regression analysis including all factors identified in a statistical comparison between the patients who lived and those who died (or the factors found in a univariate logistic regression analysis with mortality as the dependent variable to have a low p-value). If this analysis identifies quinolone use as an independent factor associated with mortality, this would be a strongly confirmatory finding. A drawback would be the small number of patients that died, which may not suffice for this type of analysis

Author Response

Response: We thank the reviewer for the kind comments and suggestions.

  1. We have replaced the abbreviation “FQ” with the word “fluoroquinolone” (lines 29 – 30) since it was mentioned only a couple of times in the current version of the abstract.
  2. The change was performed as suggested (line 35).
  3. MBL means “mucosal barrier injury”. The definition of the abbreviation has been added to the figure title “Figure 1. Incidence density of bloodstream infections (BSI) related to central venous catheter (CVC-BSI) or to mucosal barrier injury (MBL-BSI) in neutropenic hematopoietic stem cell transplantation (HSCT) patients at a reference center in São Paulo, Brazil, from 2016 to 2019”. We have revised the manuscript and all abbreviations are now explained when first introduced in the abstract, the main text, the figures, and the tables.
  4. We have added BSI mortality in Table 1 as suggested.
  5. We agree that a multivariate regression model including all confounding variables of interest would be desirable and that the identification of quinolone use as an independent risk factor for mortality in that analysis would represent a strongly confirmatory finding. However, the restricted sample size and the consequent low number of patients that died impair the inclusion of several independent variables in our regression model. This factor might even be a problem for the addition of one confounding variable at a time, as we had done (we have mentioned this limitation in the discussion, lines 171 – 174). We believe that including several independent variables would make the analysis even more imprecise than the adjustment for confounding variables in separate regression models. Therefore, we kept the regression analyses with confounding variables in separate regression models. We present the multivariate logistic regression analysis including factors found in a univariate logistic regression analysis with mortality as the dependent variable to have p-value <0.1 in a statistical comparison between the patients who lived and those who died in the attached file (Table_R).

Reviewer 3 Report

Although the interesting topic, the introduction should be improved. Please better define the design of the study and the reason why you have chosen a quasiexperimental design. 

Author Response

Response: We thank the reviewer for the kind comments and suggestions.

  • A suggested, we have reformulated the introduction:

“Bloodstream infection (BSI) is the most common serious infectious complication in patients undergoing hematopoietic stem cell transplantation (HSCT). Depending on the protocol used for transplantation and the duration of neutropenia, 13% - 62% of patients develop BSIs, which can result in increased length of hospital stay, costs, and mortality [1-3].

Changes in the care of HSCT recipients have shown to impact BSI etiology, especially regarding microorganisms resistance patterns [4]. Although fluoroquinolone (FQ) prophylaxis has been proven to reduce the rate of BSI during neutropenia, the in-creasing rate of BSI due to fluoroquinolone-resistant and extended-spectrum b-lactamase (ESBL)-producing gram-negative bacteria is a growing concern [5], and multidrug resistant gram-negative pathogens have emerged worldwide [6]. These epidemiologic trends are important to consider when choosing an empirical antibiotic treatment because inadequate coverage is associated with increased mortality [7].

We have driven the hypothesis that nonuse of universal levofloxacin prophylaxis during neutropenia could reduce antibiotic resistance in gram-negative bacteria that cause BSI in HSCT patients and that this change in resistance pattern could lead to impact on BSI mortality.”

  • The quasi-experimental design chosen was a “before-after” study. This information has been added to the manuscript (line 189). We have chosen this study design because there was a change in the institutional protocol for antibiotic prophylaxis in HSCT in our center between 2018 and 2019. Therefore, we decided to compare data before and after this change in the protocol.

Round 2

Reviewer 2 Report

The manuscript has been improved during the revision process.

Author Response

Thank you so much for your kind comments.